# How to Leverage Predictive Uncertainty Estimates for Reducing Catastrophic Forgetting in Online Continual Learning

**Giuseppe Serra**                                           *serra@med.uni-frankfurt.de*
*Goethe University Frankfurt, Frankfurt, Germany*
*German Cancer Consortium (DKTK)*, Frankfurt, Germany*

**Ben Werner**                                                    *bwerner.sci@gmail.com*
*Goethe University Frankfurt, Frankfurt, Germany*

**Florian Buettner**                            *florian.buettner@dkfz-heidelberg.de*
*Goethe University Frankfurt, Frankfurt, Germany*
*German Cancer Consortium (DKTK)*, Frankfurt, Germany*
*German Cancer Research Center (DKFZ), Heidelberg, Germany*

**Reviewed on OpenReview:** *https://openreview.net/forum?id=dczXeOS1oL*

## Abstract

Many real-world applications require machine-learning models to be able to deal with non-stationary data distributions and thus learn autonomously over an extended period of time, often in an online setting. One of the main challenges in this scenario is the so-called catastrophic forgetting (CF) for which the learning model tends to focus on the most recent tasks while experiencing predictive degradation on older ones. In the online setting, the most effective solutions employ a fixed-size memory buffer to store old samples used for replay when training on new tasks. Many approaches have been presented to tackle this problem and conflicting strategies are proposed to populate the memory. Are the easiest-to-forget or the easiest-to-remember samples more effective in combating CF? Furthermore, it is not clear how predictive uncertainty information for memory management can be leveraged in the most effective manner. Starting from the intuition that predictive uncertainty provides an idea of the samples' location in the decision space, this work presents an in-depth analysis of different uncertainty estimates and strategies for populating the memory. The investigation provides a better understanding of the characteristics data points should have for alleviating CF. Then, we propose an alternative method for estimating predictive uncertainty via the generalised variance induced by the negative log-likelihood. Finally, we demonstrate that the use of predictive uncertainty measures helps in reducing CF in different settings.

## 1 Introduction

Typical machine learning models assume to work in a single-task static scenario where multiple epochs are performed over the same data until convergence. In many real-world situations, however, this setting is too limiting. As an example, let us consider the problem of product recommendation. Since trends may change and new types of products may arrive, new product categories need to be classified. In this context, a typical learning model would fail because the standard setup does not account for the continuous addition of new classes. For this reason, *online* Continual Learning (online-CL) has been constantly more explored. In online-CL, a single model is required to learn continuously from a sequence of tasks that comes as a stream of tiny batches which can be processed only once (Aljundi et al., 2019b; 2017; Mai et al., 2022). This is to reflect

---

*partner site Frankfurt, a partnership between DKFZ and UCT Frankfurt-Marburg

realistic conditions where, for example, new personal data arrive with high-frequency to limited-resources devices (e.g., wearable smart devices) and the model needs to be updated with the incoming data to adjust the model on the fly. In this context, given the overlap between old and new information, the model tends to forget about the past knowledge (which still needs to be classified) to focus more on the newest tasks, leading to a performance degradation on previous tasks. This challenge is usually referred to as *catastrophic forgetting* (CF) (McCloskey & Cohen, 1989; Ratcliff, 1990).

Many approaches have been developed in online-CL to prevent catastrophic forgetting and the most successful ones fall in the *memory-based* category. These approaches employ a *memory buffer* (Chaudhry et al., 2019; Soutif-Cormerais et al., 2023) to a) store samples of past tasks and b) tackle CF by training the model on both current samples and some old samples stored in such limited-size memory (*replay* or *rehearsal*). In this context, what differentiates each memory-based approach are the *update* (or *population*) strategy and the *retrieval* (or *sampling*) strategy (Mai et al., 2022) — i.e., how to populate and update the memory with meaningful and representative samples, and how to efficiently sample from the memory respectively.

Although many approaches have been developed in this direction – ranging from using random approaches (Chaudhry et al., 2019) to exploiting the gradient (Lopez-Paz & Ranzato, 2017; Chaudhry et al., 2018b; Aljundi et al., 2019b) or the loss (Belilovsky et al., 2019) information – it is difficult to identify a clear strategy to exploit the memory at its best. In fact, contrasting strategies can be found in the literature. For instance, Kumari et al. (2022) suggest working on instances close to the decision boundary, thus considering the marginal samples as the most important for alleviating CF. Contrarily, in Hurtado et al. (2023) and Yoon et al. (2021), the proposed strategies focus on populating the memory with the most representative samples for each class. Hence, it is unclear which type of data points would reduce CF in a consistent manner: **Are the least or the most representative samples more effective in combating CF?**

Starting from this question, our investigation seeks to understand the contribution of the most or least representative samples in combating CF through an uncertainty lens. In detail, we will focus on the uncertainty of the model in its predictions, i.e., *predictive uncertainty*. Intuitively, using measures of uncertainty to populate the memory represents a solution for identifying the location of the instances in the decision space. However, predictive uncertainty can be seen as a composition of *aleatoric* (data-inherent and irreducible) and *epistemic* (model-centric and reducible by gathering more data) uncertainty (Malinin & Gales, 2018; Mucsányi et al., 2024), and different uncertainty scores focus more on one source or another leading to different results accordingly. The most common confidence scores mostly capture the (irreducible) *aleatoric* uncertainty (Wimmer et al., 2023) which is inherent in the data and cannot be reduced by gathering more data. In this case, samples with high confidence from an aleatoric perspective reflect data points that are distant from the decision boundary irrespective of the fact that they may be outliers or not. In the epistemic case, instead, samples with high confidence (i.e., low epistemic uncertainty) indicate that the observed data sufficiently support the inferred patterns (or, in other words, that the samples are *representative* of the data distribution). Thus, although representativeness is an intrinsic property of the data, we can interpret (low) epistemic uncertainty as a way to infer this property from a predictive model (see Section 4.2.3 for further details). For this reason, **we hypothesize it would be beneficial to focus on samples with a low *epistemic* uncertainty**. To empirically validate this hypothesis, we propose a new population strategy based on recent theoretical contributions in uncertainty quantification via the Bregman decomposition. More specifically, we introduce a memory management strategy based on the Bregman Information (BI) as a generalised variance measure, which stems directly from a bias-variance decomposition of the model loss (Gruber & Buettner, 2023). Based on this insight, we interpret BI as a measure of epistemic uncertainty that is statistically well grounded in a bias-variance decomposition, and propose to use it for populating the memory.

In the first part of this work, we evaluate and compare different combinations of uncertainty scores and sorting on CIFAR-10 and CIFAR-100 (Krizhevsky et al., 2009), two datasets commonly used in CL. The objective is to understand how different uncertainty estimates behave when used in different ways under same conditions. In order to achieve the intended goal, following a recent trend on research transparency and comparability (Mundt et al., 2021), the evaluation framework will be freed from any other 'trick' that could create ambiguity in the assessment. The investigation will focus on describing the characteristics of the considered uncertainty scores, while providing an in-depth analysis of the effect of the metrics under a memory-based regime. In

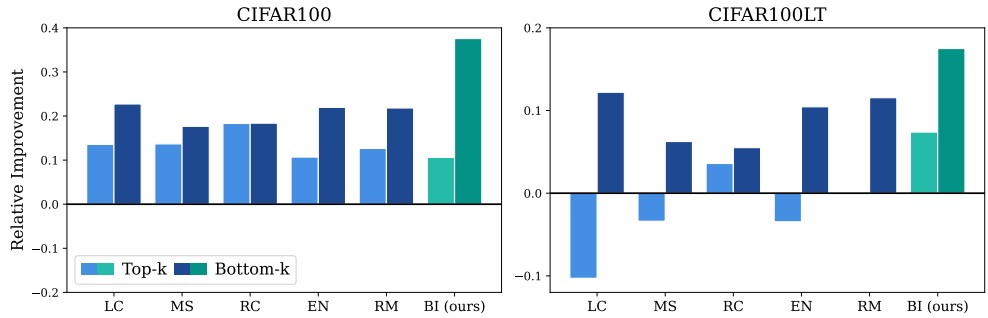

Figure 1: Relative CF improvement of different uncertainty scores over ER when using the easiest-to-forget (top-k) and the easiest-to-remember (bottom-k) samples. The bottom-k strategy (in darker colour) reduces CF in all cases. Furthermore, the proposed BI-based uncertainty estimate (in green) further diminishes CF compared to common uncertainty scores.

the second part of our work, we evaluate our findings on more challenging and realistic scenarios. In realistic setups, recent tasks have less data points available than older tasks as the time to collect instances is considerably shorter than for previous tasks. Considering this behaviour, we will focus on long-tailed (LT) datasets with this characteristic. We test our findings on two datasets with controllable degrees of data imbalance, as well as a real-world imbalanced dataset for classification of biomedical images (Yang et al., 2023). The goal of this work is not to propose a new method that outperforms the state-of-the-art, but rather to present practical insights from an uncertainty-aware perspective on the desirable characteristics samples should have to alleviate catastrophic forgetting in both standard and realistic scenarios.

Thus, the main contributions of this paper can be summarised as follows:

- We conduct a systematic evaluation of established predictive uncertainty scores to assess their effectiveness in mitigating catastrophic forgetting (CF) in online continual learning for image classification tasks.

- We propose the use of Bregman Information (BI) as measure of *epistemic* uncertainty to populate the memory in online-CL.

- As summarised in Figure 1, we demonstrate that a) selecting the most representative samples (bottom-k) to populate the memory consistently reduces CF, as compared to choosing marginal samples (top-k); and b) focusing on estimates of *epistemic* uncertainty (BI) provides an advantage compared to *aleatoric* uncertainty estimates.

- Different from conventional evaluation pipelines, we validate our findings on both standard benchmarks and realistic scenarios, including challenging settings like medical image classification with imbalanced data.

## 2 Related Work

### 2.1 Continual Learning

According to van de Ven et al. (2022), three different scenarios of incremental learning (IL) can be identified; 1) *Task-IL* where the task ID information is available at both training and testing time; 2) *Domain-IL* in which the learning task remains unchanged (e.g., binary classification) but there is a shift in the input distribution; and 3) *Class-IL* where the number of classes to discriminate can grow over time. Additionally, CL problems can be further categorised depending on whether the data can be accessed and processed multiple times (*offline*) or a single-pass through the data is expected (*online*) (Mai et al., 2022). In this work, following a popular trend in the literature, we focus on the most challenging scenario, i.e., online class-IL.

## 2.2 Class-Incremental Learning

Following the classification proposed by Mai et al. (2022), class-IL approaches can be grouped into the following categories: a) Regularization techniques that adjust the model parameter updates by incorporating penalty terms in the loss function (Aljundi et al., 2018; Lee et al., 2017), modifying parameter gradients during optimization (Chaudhry et al., 2018b; He & Jaeger, 2018), or employing knowledge distillation (Wu et al., 2019; Rannen et al., 2017); b) Memory-based techniques in which a fixed-size subset of past samples is stored for replay (Aljundi et al., 2019a; Chaudhry et al., 2019) or for regularization purposes (Nguyen et al., 2018; Tao et al., 2020) ; c) Generative-based techniques which involve training generative models to produce pseudo-samples that replicate the information from past tasks (Lesort et al., 2019; Shin et al., 2017), and d) Parameter-isolation-based techniques that allocate distinct model parameters to each task, either by activating only the relevant parameters for each task (Fixed Architecture) (Mallya & Lazebnik, 2018; Serra et al., 2018) or by adding new parameters while keeping the existing ones unchanged (Dynamic Architecture) (Yoon et al., 2018; Aljundi et al., 2017).

In online-CL, where data are received in mini-batches and the model is updated with high frequency, rehearsal-based methods are favoured over more complex solutions like generative methods due to their flexibility and reduced training time (Mai et al., 2022).

### 2.2.1 Population Strategies in Rehearsal-based CL

As already anticipated, there exist methods that use different metrics for deciding the samples to store in the memory. In Aljundi et al. (2019b), the gradient information is used as a feature to maximize the diversity of the samples in the memory. In Chaudhry et al. (2018b), the gradient of the current mini-batch is compared with with the gradient of a randomly sampled set of the same size from the memory. If the dot product between the current gradient and the memory gradient is negative, the current gradient is projected. Otherwise, the gradient is used normally. Similarly, in Yoon et al. (2021), gradient vectors are used to select the most representative and informative coreset at each iteration. Other methods exploit the loss information to select samples based on their interference on the loss function (Aljundi et al., 2019a) or to analyze the loss region for improving generalization and minimizing CF (Verwimp et al., 2021). Sun et al. (2021) and Wiewel & Yang (2021) address the problem of memory diversity from an information-based perspective via entropy-based functions. As previously mentioned, the characteristics samples should have to be stored in the memory and alleviate CF are not clear. For instance, Hurtado et al. (2023) propose a method to eliminate outliers from the memory such that only the most representative samples for each class are kept via a label-homogeneity score. Intuitively, their approach inspect the neighborhood of a given sample to check whether it is surrounded by samples from different classes or not. If the label homogeneity score is low (i.e., most of the nearest samples do not share the same class label), it means we are in the presence of an outlier or noisy sample and the considered sample should be discarded from memory sampling. Contrarily, in Kumari et al. (2022), the authors claim that the replay-phase should focus on *marginal samples* and propose a method to synthesize samples near the forgetting boundary that are confused with the current task. The approach consists on identifying the easiest-to-forget samples in the memory and then on moving those samples closer to the data points of the current task. Finally, the new perturbed samples are used for replay to cover marginal samples more consistently.

In both cases, the idea is similar to using uncertainty estimates to detect data points with specific characteristics – i.e., outliers or representative samples. Differently from their approaches, which require the samples' vector representation, our method exploits predictive uncertainty for the same purpose. It can be seen as a more principled way of identifying whether a sample is representative of its class or not. Nevertheless, despite the use of predictive uncertainty looks appealing, its usage for populating the memory with samples with desired properties is largely unexplored.

### 2.2.2 Uncertainty-aware Memory Management in CL

The idea of employing predictive uncertainty for memory management is borrowed from Active Learning (AL). In this context, uncertainty sampling (Lewis & Gale, 1994) is used to decide which samples from a pool of unlabeled data will be considered for labeling. The main assumption is that the most uncertain

samples represent the most informative data points. Thus, by including them in the training set, we can improve the overall performance. Popular examples of uncertainty scores include, to name a few, entropy, smallest margin, and least confidence (Shannon, 1948; Campbell et al., 2000; Lewis & Gale, 1994; Culotta & McCallum, 2005).

Analogously, we can use estimates of predictive uncertainty for populating memory buffers in replay-based methods for online-CL. In Bang et al. (2021), the authors argue that samples stored in the memory should be representative of their own class, but also discriminative towards the remaining ones. Starting from this assumption, they evaluate the relative position of the samples in the decision space by estimating the predictive uncertainty of perturbed samples; exemplars with high uncertainty should be closer to the decision boundary, while the most certain ones should be located closer to the center of the corresponding class distribution. To ensure diversity, they use a step-sized sampling which guarantees to keep in the memory samples that span from the most uncertain to the most certain ones. Given a list $P$ of perturbations, the authors propose to use the following uncertainty score $u_{RM}(x)$:

$$u_{RM}(x) = 1 - \frac{1}{P} \max_c S_c \tag{1}$$

where $S_c = \sum_{i=1}^{P} \mathbb{1}_c \operatorname{argmax}_{\hat{c}} p(y = \hat{c}|\tilde{x}_i)$. Eq. (1) represents an agreement score with respect to the perturbations. If the predicted classes $\hat{c}$ of all the perturbed versions of $x$ (i.e., $\tilde{x}_i$) are predicted with the same label $c$, then $u_{RM}(x)$ will be 0. Otherwise, the higher is $u_{RM}(x)$ the most uncertain is the model about the considered sample.

The authors claim to work in the online setting. Nevertheless, the definition of online setting in the paper differs from the popular one. In their case, *online* means that the training stream is only processed once and the memory is updated at the end of each task. In the standard scenario, instead, *online* implies that only a mini-batch of data points is available to be processed (Mai et al., 2022), and the model and the memory are updated with high frequency (Soutif-Cormerais et al., 2023). In addition to this, to further enhance diversity, they employ data augmentation on the memory. Thus, the contribution of Eq. (1) on reducing CF is unclear.

In the remainder of this work, we systematically investigate the effect of established uncertainty scores - including $u_{RM}$ - as well as the newly proposed Bregman Information on combating catastrophic forgetting in a realistic online-CL setting.

## 3 Preliminaries and Notation

Following the notation proposed in Bang et al. (2021), we assume to have a set $\mathcal{C} = \{c_1, \ldots, c_n\}$ of $n$ different classes. Each class can be randomly assigned to a task $t$. $T_t$ represents a subset of classes determined by an assign function $\psi(c)$ such that $T_t = \{c|\psi(c) = t\}$. For each task $t$, we have an associated dataset $D_t = \{(x_i, y_i)\}_{i=1}^{n_t}$ with $x_i$ an input sample, $y_i$ the corresponding class label, and $n_t$ the number of training samples. In the proposed *online* setting, we assume that samples for each task $t$ arrive sequentially in a stream of mini-batches $b_t = \{(x_i, y_i) \in D_t\}_{i=1}^{bs}$, with each mini-batch available for a single pass. It is important to notice that, for each task $t$, $n_t$ can vary depending on whether we are working on a class-balanced setting or not. Finally, for replay, we introduce a fixed-size memory buffer $\mathcal{M}$ to store a portion of samples from past tasks. The memory is updated with high-frequency whenever a mini-batch from the stream of data is processed. Following the standard procedure in online-CL, for replay, we assume to extract from the memory a number of samples equal in size to the batch size.

## 4 Methodology

To facilitate a fair evaluation of the framework, we focus on the assessment of the uncertainty metrics under same conditions. In this way, we eliminate any ambiguous effect which could be inherited from other methodological choices. As anticipated in Section 1, there are two main objectives when dealing with memory-based approaches: 1) memory populating, and 2) memory sampling.

In the following subsections, we describe the available strategies for each of the two objectives, and the uncertainty scores considered in our evaluation. Finally, we propose an alternative uncertainty estimate that overcomes the limitations of the most popular uncertainty metrics.

## 4.1 Memory Management

**Memory Population.** Apart from the criteria-based population strategies presented in Section 2.2.1, random strategies have demonstrated to work surprisingly well in online-CL. *Reservoir* (Vitter, 1985) is a random sampling technique without replacement to select $n$ samples from a pool of $N$ samples where $N$ is unknown and $N > n$. This strategy is used in Chaudhry et al. (2019) as a way to populate a memory of size $M$ from a stream of data points with unknown length. In this way, each data point has a probability of being included in the memory equal to $\frac{M}{N}$. *Class-balanced Reservoir*, instead, is based on a class-balanced random sampling strategy (Chrysakis & Moens, 2020). This is important to consider the case where the stream of data is highly imbalanced. Indeed, class-imbalance may further deteriorate the predictive performance of the framework.

Different from the population strategy proposed in Chrysakis & Moens (2020), where each class-memory set is populated and updated randomly, we introduce a class-balanced memory management based on predictive uncertainty estimates. This is similar to Bang et al. (2021), where the intent is to keep in the memory samples with desired characteristics for each class.

**Memory Sampling.** Starting from the second task, we need a strategy to sample data points from the memory. The easiest way is *random sampling*, which selects at random a subset from the available data points in the memory buffer excluding the ones from the current task. The size of the replay set is equal to the batch size. Another option would be to extract past samples with specific characteristics. For this purpose, we can exploit uncertainty estimates. However, this solution represents an effective strategy only in the case the memory is populated at random. Furthermore, given the size of the memory buffer and the frequency of memory updates, computing uncertainty estimates for the whole buffer may be computationally expensive and not feasible during training.

In our case, since the memory is populated with samples with specific characteristics, the random sampling is sufficient.

## 4.2 Uncertainty Metrics

### 4.2.1 Predictive Uncertainty Estimation

Following previous works (Bang et al., 2021; Wang et al., 2022), to compute predictive uncertainty estimations for a given input image $x$, we employ the ensembling technique Test-Time Augmentation (TTA) (Wang et al., 2019). The set of transformations applied are the ones usually employed for image classification. A detailed list of the transformations used in our evaluation can be found in the supplementary. After selecting the set of transformations, we apply them on the selected input. In this way, we create a set of $P$ perturbed inputs $\tilde{x}$ which are used at test-time to compute their corresponding logits $\hat{z}$. Finally, the generated logits are used by the uncertainty estimates in different ways to compute the uncertainty score $u(x)$.

### 4.2.2 Predictive Uncertainty Scores

In our investigation, we consider the following popular uncertainty metrics:

- *Least Confidence (LC)* (Culotta & McCallum, 2005) determines the level of predictive uncertainty by examining the samples with the smallest predicted class probability. A model is less certain about the sample if it associates the most probable class $y_{(1)}$ with a low probability.

$$u(x)_{LC} = 1 - \frac{1}{P} \sum_{i=1}^{P} p(y_{(1)} = \hat{c}|\tilde{x}_i) \tag{2}$$

- *Margin Sampling (MS)* (Campbell et al., 2000) calculates predictive uncertainty by computing the difference between the most probable predicted class $y_{(1)}$ and the second largest one $y_{(2)}$. If the difference is low, the model is uncertain.

$$u(x)_{MS} = 1 - \frac{1}{P} \sum_{i=1}^{P} \left( p(y_{(1)} = \hat{c}|\tilde{x}_i) - p(y_{(2)} = \hat{c}|\tilde{x}_i) \right) \tag{3}$$

- *Ratio of Confidence (RC)* (Campbell et al., 2000) is similar to MS. In this case, the ratio between the probabilities of the two most probable classes is computed. The closer the ratio is to 1, the more uncertain the model is about the considered sample.

$$u(x)_{RC} = \frac{1}{P} \sum_{i=1}^{P} \left( \frac{p(y_{(2)} = \hat{c}|\tilde{x}_i))}{p(y_{(1)} = \hat{c}|\tilde{x}_i))} \right) \tag{4}$$

- *Entropy (EN)* (Shannon, 1948), differently from the above scores, estimates the uncertainty considering the whole probability distribution. If the predicted probabilities are similar to each other (i.e., tend to an uniform distribution), the model is not certain about the sample. The formulation reads as follows:

$$u(x)_{EN} = -\frac{1}{P} \sum_{i=1}^{P} \left( \sum_{j} p(y_j = \hat{c}|\tilde{x}_i) \log(p(y_j = \hat{c}|\tilde{x}_i)) \right) \tag{5}$$

In addition to these metrics, we will also consider the agreement score $u(x)_{RM}$ reported in Eq. (1).

### 4.2.3 Uncertainty Quantification via Bregman Decomposition

Inspecting the metrics described above, we can make two main observations: 1) all the metrics need a normalization step to squash the logits in the $[0, 1]$ range of the probability space; 2) the majority of them compute the uncertainty relying on the largest predicted probabilities only. In both cases, with different magnitudes, there is a loss of information compared to using the logit space. Importantly, such confidence scores are a reliable measure of predictive uncertainty only when the model is well calibrated, which is not the case in many real-world scenarios (Gruber & Buettner, 2023; Ovadia et al., 2019; Tomani & Buettner, 2021).

Furthermore, the confidence scores detailed in Section 4.2.2 mostly capture the aleatoric uncertainty which is inherent in the data and *irreducible* (Wimmer et al., 2023). For this reason, we believe it would be favorable to focus on samples with low *epistemic* uncertainty which is model-centric and can be reduced by gathering more data. To provide an intuitive explanation behind this decision, let us consider Figure 2 where we have a graphical representation of the behaviour of the two different sources of uncertainty. With commonly used confidence scores, regions with low confidence correspond to areas close to the decision boundary irrespective of the presence or not of outliers. Contrarily, when exploiting measures mostly based on epistemic uncertainty, low confidence regions correspond to areas in which the data density is low, i.e., where the uncertainty could be reduced by gathering more data. Consequently, if we are interested in the high confidence samples, this would result in selecting different samples depending on the employed score. For confidence scores, given their distance from the decision boundary, we may obtain outliers not being representative of the class of interest.

In light of these considerations, following recent advances for computing uncertainty estimates in the logit space for classification tasks, we suggest to employ an uncertainty estimator based on the Bregman Information (BI) (Gruber & Buettner, 2023). The authors demonstrate that BI can be used to quantify the variance of the model at the sample level through deep ensembles or TTA. This approach is particularly appealing from a theoretical standpoint, since it stems from a general bias-variance decomposition of proper scoring rules which gives rise to the Bregman Information as the variance term of the loss.

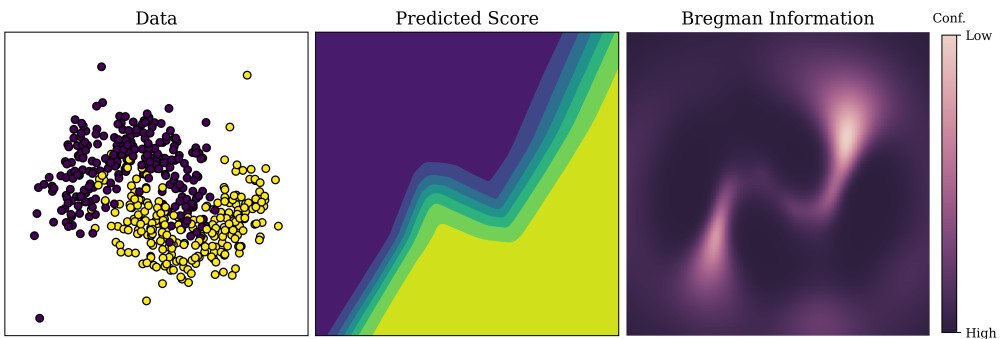

Figure 2: Illustration of the behaviour of the Bregman Information for uncertainty estimation in classification tasks. Given the focus on epistemic uncertainty, BI is low in the presence of high data density areas – i.e., where the uncertainty could be reduced by gathering more data.

For the commonly used cross-entropy loss, this variance term – that can be directly associated to the epistemic uncertainty – can be estimated via $P$ TTA perturbations as follows:

$$u(x)_{BI} = \frac{1}{P} \sum_{i=1}^{P} \text{LSE}(\hat{z}_i) - \text{LSE}\left(\frac{1}{P} \sum_{i=1}^{p} \hat{z}_i\right), \tag{6}$$

where $\hat{z}_i \in \mathbb{R}^c$ and $\text{LSE}(x_1, \ldots, x_n) = \ln \sum_{i=1}^{n} e^{x_i}$ represent the logits generated by the model and the *LogSumExp* (LSE) function respectively. Intuitively, a large value of $u(x)_{BI}$ means that the logits predicted across the perturbations are very different and, thus, the model is not confident about the considered sample.

### 4.2.4 Uncertainty Scores Ranking

Depending on the characteristics we want to extract from the images, we can use the uncertainty estimates in different ways. In particular, in our experiments we consider three different strategies to select the set of images for populating the memory depending on the uncertainty scores. By sampling images with the highest uncertainty (*top-k*), we aim at extracting the easiest-to-forget samples which are closer to the decision boundaries and/or represent outliers. With the *step-sized* strategy, we aim at sampling a diverse set of images, spanning from the most representative to the most uncertain ones. Finally, with the *bottom-k* approach we aim at finding the most representative set of images for each of the classes seen in the past tasks. Note that, as explained above, different uncertainty scores select different images based on their greater focus on epistemic or aleatoric uncertainty.

## 5   Experiments

### 5.1   Datasets and Settings

#### 5.1.1   Datasets

To understand the change in the performance when using different uncertainty strategies, we employ two datasets commonly used in online-CL for image classification, CIFAR-10 and CIFAR-100 (Krizhevsky et al., 2009). To configure the online-CIL setup, we randomly assign with different random seeds a set of classes to 5 tasks. Thus, each task $T_t$ has 2 (CIFAR-10) or 20 (CIFAR-100) classes designated. In the second part of our experiments, once the best strategy is identified, we evaluate our findings on class-imbalanced scenarios to assess the performance under more realistic conditions. First, we employ two artificially controlled imbalanced datasets, CIFAR10-LT and CIFAR100-LT (Cao et al., 2019). As explained in Section 1, with this experiment we want to replicate a realistic scenario in which recent tasks contain less data than the older ones. For this, the most appropriate type of imbalance is the long-tailed (LT) one (Cui et al., 2019)

which follows an exponential decay to choose the sample size of each class. Specifically, for both datasets, we decided to use an imbalance factor $\rho$ equal to 0.1. This means that, if the largest class contains, e.g., 500 data points, then the smallest one consists of 50 instances. Finally, we decide to focus on biomedical image analysis using the microscopic peripheral blood cell images dataset (BloodCell) which consists of 8 classes annotated by expert clinical pathologists (Acevedo et al., 2020; Yang et al., 2023). To reflect the intended realistic conditions, we assign 2 classes for each tasks by increasing size. In this way, older tasks have more data points than the most recent ones. This simulates a realistic scenario where most common subtypes are prevalent when collecting data for a ML-model and, over time, less common disease subtypes appear in the clinic and the model needs to be updated with the new incoming classes. We provide more details about the motivations behind the choice of this dataset in Appendix A.4.

### 5.1.2 Experimental Settings

The baseline for assessing the goodness of the proposed strategies will be Experience Replay (ER) (Chaudhry et al., 2019). ER consists of a reservoir approach for the memory management part, and a random sampling for replay. We decide to use ER because, despite its simplicity, it is surprisingly competitive compared to more sophisticated and newer approaches. This is corroborated by recent empirical surveys finding that newly introduced methods perform very similarly to the common Experience Replay (ER) method (Soutif-Cormerais et al., 2023). In addition to the standard strategy, we will also consider the class-balanced version (CBR) proposed in Chrysakis & Moens (2020) and previously described in Section 4.1, and a gradient-based method (A-GEM Chaudhry et al. (2018b)) for comparison. Finally, we assess the results in comparison with Monte Carlo Dropout (MC) (Gal & Ghahramani, 2016). As reported in Ovadia et al. (2019), MC can be used to estimate predictive uncertainty and considered as a competitive baseline under distribution shifts. Since online-CL can be seen as a special case of data under distribution shifts, we include MC as an additional baseline to estimate predictive uncertainty from a Bayesian perspective.

In all the experiments, we employ a slim version of Resnet18 (He et al., 2016) – as done in previous works (Hurtado et al., 2023; Lopez-Paz & Ranzato, 2017; Soutif-Cormerais et al., 2023; Kumari et al., 2022) –, and use the SGD optimizer with a learning rate of 0.1. Following standard practice, we set the batch size equal to 10. We set the memory size to different values for each dataset to evaluate a variety of memory configurations considering both large or small buffers.

For evaluation, following previous works, we employ the *Last Accuracy* (A) and *Last Forgetting* (F) – defined in Chaudhry et al. (2018a). *Last* refers to the calculation of the metrics at the end of the training on all tasks. Let denote with $a^{t,i}$ and $T$, the accuracy of task $i$ after learning task $t$ and the total number of tasks respectively.

The last accuracy $A$ is then defined as follows:

$$A = \frac{1}{T} \sum_{i=1}^{T} a^{T,i}. \tag{7}$$

*Last Forgetting* (F) is defined as the difference between the peak knowledge (i.e., the maximum accuracy) captured about a particular task during the learning process and its accuracy at the end of the learning process. This provides an idea of the information retained about a certain task after learning new tasks. The last forgetting (F) is computed as:

$$F = \frac{1}{T-1} \sum_{t=1}^{T-1} \max_{l \in \{1,\dots,T-1\}} a^{l,j} - a^{T,j}, \qquad \forall j < T. \tag{8}$$

All the experiments are run on three different random seeds. The code to conduct the experiments is available at `https://github.com/MLO-lab/uncertainty_estimates_for_CF`.

Table 1: Comparison of last accuracy (A) and last forgetting (F) on CIFAR10.

| Score | | M=500 | | | M=1000 | | |
|---|---|---|---|---|---|---|---|
| ER | | 26.38 ± 2.27 74.92 ± 4.93 | | | 36.50 ± 2.16 57.42 ± 5.93 | | |
| CBR | | 24.65 ± 2.10 76.70 ± 5.29 | | | 30.88 ± 1.89 66.40 ± 5.58 | | |
| A-GEM | | 28.81 ± 1.36 56.56 ± 0.18 | | | 30.55 ± 1.54 56.02 ± 4.49 | | |
| | | Top | Step | Bottom | Top | Step | Bottom |
| LC | A(↑) | 25.27 ± 0.93 | 30.51 ± 3.32 | 34.49 ± 2.91 | 30.70 ± 2.22 | 38.03 ± 2.98 | 38.30 ± 2.98 |
| | F(↓) | 72.88 ± 2.76 | 58.07 ± 6.23 | 53.11 ± 5.85 | 64.48 ± 4.09 | 48.04 ± 6.54 | 41.71 ± 5.68 |
| MS | A(↑) | 26.33 ± 2.07 | 30.16 ± 0.99 | 32.50 ± 2.31 | 33.29 ± 1.41 | 38.85 ± 4.31 | 39.36 ± 1.39 |
| | F(↓) | 71.08 ± 5.00 | 61.19 ± 5.74 | 56.03 ± 4.50 | 59.75 ± 1.87 | 48.89 ± 8.57 | 43.84 ± 3.42 |
| RC | A(↑) | 27.84 ± 2.33 | 30.57 ± 2.43 | 32.49 ± 3.86 | 36.35 ± 1.35 | 37.68 ± 1.17 | 39.63 ± 0.80 |
| | F(↓) | 66.13 ± 7.78 | 59.70 ± 6.71 | 54.73 ± 7.66 | 54.53 ± 4.54 | 48.08 ± 3.69 | 41.9 ± 3.53 |
| EN | A(↑) | 23.72 ± 1.50 | 28.76 ± 1.76 | 35.20 ± 1.52 | 26.57 ± 0.98 | 36.13 ± 1.56 | 35.78 ± 4.99 |
| | F(↓) | 77.04 ± 3.78 | 61.20 ± 3.33 | 46.12 ± 3.02 | 71.66 ± 3.78 | 50.37 ± 3.44 | 42.65 ± 8.07 |
| RM | A(↑) | 26.92 ± 1.75 | 28.57 ± 0.80 | **35.71 ± 4.12** | 32.87 ± 2.56 | 38.23 ± 0.95 | 38.57 ± 2.09 |
| | F(↓) | 71.45 ± 2.92 | 61.16 ± 4.40 | 51.01 ± 6.60 | 62.01 ± 4.36 | 47.21 ± 3.68 | 42.59 ± 2.79 |
| MC | A(↑) | 25.28 ± 2.05 | 28.72 ± 2.57 | 27.52 ± 4.84 | 28.96 ± 2.69 | 32.87 ± 3.31 | 28.04 ± 1.58 |
| | F(↓) | 44.97 ± 5.56 | 47.05 ± 5.57 | **38.01 ± 3.38** | 45.10 ± 3.13 | 39.18 ± 6.23 | 40.85 ± 5.87 |
| BI | A(↑) | 26.41 ± 1.17 | 28.88 ± 1.96 | 34.10 ± 6.24 | 28.62 ± 2.14 | 37.44 ± 2.87 | **40.44 ± 3.09** |
| | F(↓) | 72.68 ± 3.81 | 61.21 ± 6.12 | 43.03 ± 5.65 | 69.02 ± 4.78 | 47.68 ± 5.96 | **37.79 ± 6.07** |

Table 2: Comparison of last accuracy (A) and last forgetting (F) on CIFAR100.

| Score | | M=1000 | | | M=2000 | | |
|---|---|---|---|---|---|---|---|
| ER | | **14.23 ± 1.00** 29.32 ± 0.15 | | | 16.84 ± 0.96 22.67 ± 1.35 | | |
| CBR | | 13.38 ± 0.58 31.66 ± 0.30 | | | 16.59 ± 0.41 23.05 ± 1.29 | | |
| A-GEM | | 14.06 ± 0.46 33.23 ± 1.81 | | | 16.52 ± 1.05 28.80 ± 0.79 | | |
| | | Top | Step | Bottom | Top | Step | Bottom |
| LC | A(↑) | 13.33 ± 0.16 | 12.52 ± 0.65 | 14.07 ± 0.36 | 15.63 ± 1.28 | 15.06 ± 0.69 | 15.44 ± 1.31 |
| | F(↓) | 25.18 ± 1.51 | 27.50 ± 1.02 | 22.67 ± 3.02 | 19.71 ± 2.82 | 20.56 ± 1.93 | 17.50 ± 3.48 |
| MS | A(↑) | 13.69 ± 1.25 | 11.69 ± 1.07 | 13.34 ± 0.22 | 15.76 ± 0.93 | 14.73 ± 0.92 | 16.32 ± 1.11 |
| | F(↓) | 24.45 ± 1.82 | 27.05 ± 1.20 | 25.64 ± 3.62 | 20.23 ± 2.76 | 18.14 ± 1.26 | 17.50 ± 3.81 |
| RC | A(↑) | 13.65 ± 0.97 | 12.65 ± 0.07 | 13.94 ± 0.78 | 16.69 ± 1.77 | 14.30 ± 0.80 | 16.03 ± 0.88 |
| | F(↓) | 25.71 ± 2.99 | 27.55 ± 0.48 | 24.99 ± 3.48 | 17.15 ± 3.25 | 20.72 ± 1.50 | 17.68 ± 2.86 |
| EN | A(↑) | 13.00 ± 0.64 | 12.79 ± 0.30 | 13.42 ± 0.87 | 14.18 ± 1.77 | 14.49 ± 0.73 | 14.42 ± 0.78 |
| | F(↓) | 25.30 ± 2.25 | 27.82 ± 1.80 | 23.20 ± 4.00 | 20.93 ± 3.83 | 19.59 ± 2.58 | 17.43 ± 2.61 |
| RM | A(↑) | 12.99 ± 1.35 | 12.24 ± 0.22 | 13.65 ± 0.75 | 15.62 ± 0.99 | 14.33 ± 0.64 | 16.39 ± 0.52 |
| | F(↓) | 26.01 ± 3.18 | 28.08 ± 0.98 | 24.27 ± 3.52 | 19.49 ± 4.25 | 19.25 ± 1.55 | 16.68 ± 3.67 |
| MC | A(↑) | 9.48 ± 1.92 | 10.49 ± 1.05 | 10.77 ± 2.71 | 10.99 ± 1.16 | 12.36 ± 1.64 | 10.68 ± 2.75 |
| | F(↓) | 16.77 ± 2.42 | 15.09 ± 1.41 | **7.50 ± 2.53** | 15.98 ± 1.45 | 11.91 ± 0.90 | **9.08 ± 4.36** |
| BI | A(↑) | 13.38 ± 0.59 | 12.27 ± 0.69 | 13.81 ± 1.00 | 14.52 ± 0.54 | 14.33 ± 1.41 | **17.03 ± 0.30** |
| | F(↓) | 25.43 ± 3.23 | 25.50 ± 1.15 | 19.41 ± 1.74 | 20.87 ± 3.51 | 17.68 ± 1.63 | 13.30 ± 3.65 |

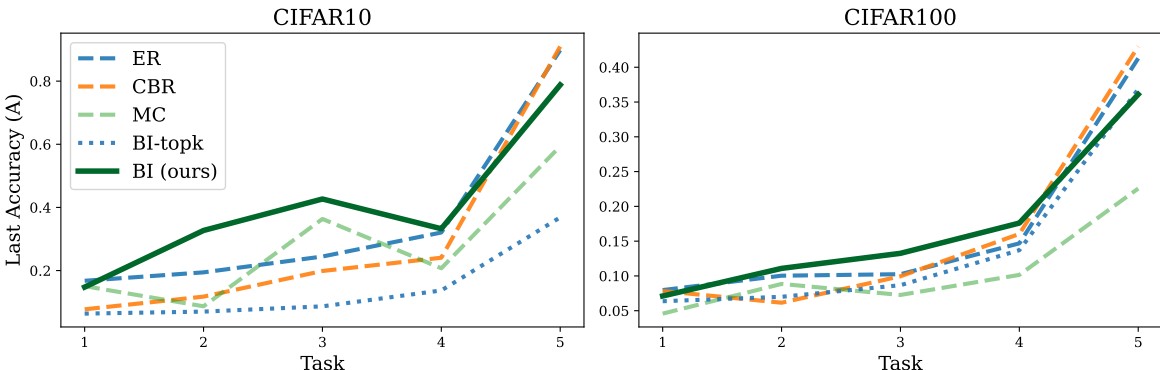

Figure 3: Average task accuracy at the end of the learning procedure for different strategies and datasets. One can observe how the proposed strategy based on the BI perform consistently across tasks and helps maintaining competitive performance on old tasks and thus reducing catastrophic forgetting.

## 5.2 Empirical Results

Tables 1 and 2 show the results for all the uncertainty metrics and sorting strategies employed for populating the memory. From the values reported in the tables, we can observe that selecting the most representative samples for each class ('Bottom' column, in green) consistently improves the results in terms of both accuracy (A) and forgetting (F) across all the considered uncertainty scores and memory sizes. In some cases, ER is able to outperform our proposed approach in terms of predictive accuracy. However, taking a closer look at the single numbers, we can observe that the higher accuracy is not accompanied by a smaller forgetting value. Thus, our results indicate that the *most representative* samples are beneficial in improving the predictive performance of the learning model and in reducing considerably CF.

The comparison among the different uncertainty strategies suggests that BI outperforms the other metrics in most cases. While the difference among the accuracy values is less pronounced, BI stands out in its effectiveness at mitigating CF. Given the reduced information loss compared to the most popular uncertainty metrics, we think that BI is more sensitive to changes in the predictive space and thus able to detect changes when subtle differences are present. We can notice that MC delivers the smallest forgetting value (F) in most of the cases. However, this is because MC performs poorly in the predictive task (see Figure 3) and considering how forgetting is computed, the resulting value is inevitably smaller. The poor performance of MC compared to other strategies is probably given by the online nature of the setting since, as reported in the original paper, dropout takes longer to converge (Gal & Ghahramani, 2016). Thus, a single-epoch setup may be insufficient for training convergence. All these findings are evident in Figure 3 where we can observe that BI is able to maintain competitive predictive performance on all tasks at the end of the training compared to other strategies.

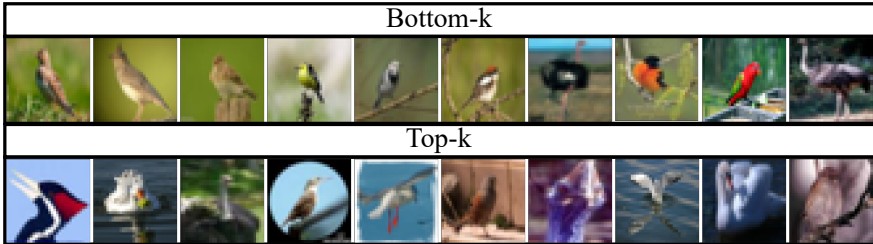

Figure 4: Class-specific samples stored in the memory according to different sorting strategies: *bottom-k* selects prototypical examples of the considered class facilitating the recollection of the class characteristics and reducing CF; *top-k* selects the most uncertain and hard samples making difficult to learn clear discriminative patterns and to reduce CF.

Table 3: Comparison of last accuracy (A) and last forgetting (F) on long-tailed imbalanced datasets.

| Score | | CIFAR10-LT | | CIFAR100-LT | | BloodCell | |
|---|---|---|---|---|---|---|---|
| | | M=200 | M=500 | M=500 | M=1000 | M=40 | M=160 |
| ER | A($\uparrow$) | $22.06 \pm 1.13$ | $31.62 \pm 1.65$ | $8.22 \pm 0.36$ | $9.30 \pm 0.22$ | $\mathbf{58.82} \pm \mathbf{6.62}$ | $64.08 \pm 11.98$ |
| | F($\downarrow$) | $76.19 \pm 3.17$ | $52.09 \pm 3.21$ | $20.16 \pm 0.9$ | $15.78 \pm 0.62$ | $49.60 \pm 8.23$ | $28.50 \pm 5.75$ |
| CBR | A($\uparrow$) | $22.38 \pm 2.84$ | $27.34 \pm 5.84$ | $8.10 \pm 0.70$ | $8.75 \pm 0.33$ | $55.74 \pm 2.24$ | $70.11 \pm 2.66$ |
| | F($\downarrow$) | $77.41 \pm 5.76$ | $62.59 \pm 7.66$ | $21.82 \pm 1.98$ | $18.32 \pm 1.84$ | $57.53 \pm 2.76$ | $29.56 \pm 5.80$ |
| A-GEM | A($\uparrow$) | $21.79 \pm 1.30$ | $23.10 \pm 1.69$ | $8.39 \pm 1.65$ | $9.95 \pm 1.83$ | $49.44 \pm 3.92$ | $55.24 \pm 3.37$ |
| | F($\downarrow$) | $63.75 \pm 6.20$ | $68.19 \pm 1.38$ | $28.60 \pm 0.96$ | $25.76 \pm 0.84$ | $54.89 \pm 4.37$ | $24.96 \pm 5.96$ |
| LC | A($\uparrow$) | $31.16 \pm 1.24$ | $32.5 \pm 1.05$ | $7.15 \pm 0.73$ | $8.93 \pm 1.05$ | $51.19 \pm 5.49$ | $66.08 \pm 1.78$ |
| | F($\downarrow$) | $42.31 \pm 2.81$ | $37.59 \pm 2.01$ | $17.53 \pm 1.71$ | $13.98 \pm 2.66$ | $27.33 \pm 0.64$ | $17.03 \pm 3.05$ |
| MS | A($\uparrow$) | $29.62 \pm 1.70$ | $33.48 \pm 1.23$ | $7.20 \pm 0.57$ | $9.11 \pm 0.66$ | $55.28 \pm 5.04$ | $68.34 \pm 3.38$ |
| | F($\downarrow$) | $47.31 \pm 2.64$ | $39.46 \pm 2.45$ | $19.36 \pm 1.79$ | $14.43 \pm 1.85$ | $34.41 \pm 5.90$ | $14.21 \pm 2.80$ |
| RC | A($\uparrow$) | $31.09 \pm 0.79$ | $32.48 \pm 1.21$ | $7.86 \pm 0.45$ | $9.47 \pm 0.59$ | $51.88 \pm 3.59$ | $63.92 \pm 4.31$ |
| | F($\downarrow$) | $44.38 \pm 1.21$ | $40.65 \pm 1.75$ | $18.51 \pm 1.38$ | $15.33 \pm 2.04$ | $28.43 \pm 2.91$ | $21.20 \pm 7.00$ |
| EN | A($\uparrow$) | $\mathbf{31.34} \pm \mathbf{0.49}$ | $32.53 \pm 1.89$ | $7.02 \pm 0.49$ | $8.48 \pm 0.71$ | $49.32 \pm 3.86$ | $69.38 \pm 1.13$ |
| | F($\downarrow$) | $41.98 \pm 2.06$ | $32.70 \pm 3.08$ | $17.79 \pm 2.18$ | $14.34 \pm 2.99$ | $32.37 \pm 2.22$ | $11.41 \pm 1.36$ |
| RM | A($\uparrow$) | $28.50 \pm 2.04$ | $33.98 \pm 2.75$ | $7.89 \pm 0.90$ | $9.90 \pm 1.12$ | $57.13 \pm 3.18$ | $67.97 \pm 4.19$ |
| | F($\downarrow$) | $51.20 \pm 5.82$ | $39.17 \pm 3.95$ | $18.36 \pm 1.27$ | $13.54 \pm 1.64$ | $31.93 \pm 4.18$ | $14.29 \pm 7.72$ |
| MC | A($\uparrow$) | $25.84 \pm 4.55$ | $23.16 \pm 1.03$ | $7.42 \pm 2.12$ | $7.76 \pm 2.23$ | $55.90 \pm 0.18$ | $51.90 \pm 5.25$ |
| | F($\downarrow$) | $43.03 \pm 10.72$ | $\mathbf{28.75} \pm \mathbf{1.55}$ | $\mathbf{7.84} \pm \mathbf{2.62}$ | $\mathbf{8.74} \pm \mathbf{2.69}$ | $\mathbf{16.96} \pm \mathbf{0.61}$ | $9.9 \pm 6.06$ |
| BI | A($\uparrow$) | $30.35 \pm 1.01$ | $\mathbf{34.37} \pm \mathbf{1.26}$ | $\mathbf{8.93} \pm \mathbf{1.02}$ | $\mathbf{10.35} \pm \mathbf{0.58}$ | $58.08 \pm 7.08$ | $\mathbf{71.88} \pm \mathbf{2.33}$ |
| | F($\downarrow$) | $\mathbf{40.66} \pm \mathbf{1.06}$ | $34.93 \pm 0.50$ | $16.31 \pm 1.48$ | $13.27 \pm 1.10$ | $18.27 \pm 7.25$ | $\mathbf{8.71} \pm \mathbf{5.00}$ |

To provide a graphical intuition of the differences between the sorting strategies and their plausible effect on the learning capabilities of the model, Figure 4 depicts a subset of class-specific samples stored in the memory at the end of the training for the two opposite population strategies, i.e. bottom-k and top-k. By inspecting the images, we note that the ones selected via the bottom-k strategy show some consistency and are easily identifiable as birds. On the opposite side, the top-k samples present images with a different perspective, zooming, or background making it more difficult, even for the human eye, to identify a bird. For this reason, we believe that the easiest-to-remember images are more useful to the model for recalling the past information while the easiest-to-forget data points may hamper the generalization of the model – since we could be in the presence of noisy images or outliers.

Finally, in Table 3 we evaluate our findings in the more realistic long-tailed data imbalance scenario. In both cases (artificial and real data imbalance), the results support our findings from the standard balanced setting: BI substantially reduces CF while delivering competitive or improved predictive performance.

## 6  Conclusion

Given the conflicting nature of the strategies available in the literature for populating the memory, our investigation seeks – under an uncertainty lens – to a) clarify which characteristics of the samples alleviate CF in a consistent manner, and b) assist practitioners in selecting the appropriate memory management strategy. Starting from the examination of the properties and behaviours of popular uncertainty estimates, we identify that they mostly capture the *irreducible* aleatoric uncertainty and hypothesize that a better strategy should focus on the *epistemic* uncertainty instead. For this, we propose to use an uncertainty estimate based on the Bregman Information which, via a general bias-variance decomposition for strictly proper scores, identifies the variance term as an estimate for the epistemic uncertainty. Our findings indicate that the most representative samples – i.e., the easiest-to-remember or the least uncertain ones – are the

most effective at mitigating CF (Figure 3) while maintaining competitive performance in classification tasks (Tables 1 and 2). Furthermore, the newly proposed BI-based uncertainty score for memory management shows its superiority in reducing CF compared to well-established uncertainty scores and other memory-based strategies (Figure 1). Due to its emphasis on epistemic uncertainty, the memory contains samples that are representative of their respective class distributions (Figure 4) while being close enough to the decision boundary, helping the learning model to distinctly shape the boundaries between the classes during training. The results on various realistic setups (Table 3) – not addressed in related work – further support our findings and demonstrate that the use of BI-based predictive uncertainty estimates for reducing CF in online-CL is appealing and well-motivated even in more challenging scenarios.

### Acknowledgments

This work was supported by the The Federal Ministry for Economic Affairs and Climate Action of Germany (BMWK, Project OpenFLAAS 01MD23001E). Co-funded by the European Union (ERC, TAIPO, 101088594). Views and opinions expressed are however those of the authors only and do not necessarily reflect those of the European Union or the European Research Council. Neither the European Union nor the granting authority can be held responsible for them.

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

# A Appendix

## A.1 List of Augmentations

The list of augmentations used for estimating the variance of the classification loss via TTA is reported in Figure 5.

```
transform_cands = [
    CutoutAfterToTensor(args, 1, 10),
    CutoutAfterToTensor(args, 1, 20),
    v2.RandomHorizontalFlip(),
    v2.RandomVerticalFlip(),
    v2.RandomRotation(degrees=10),
    v2.RandomRotation(45),
    v2.RandomRotation(90),
    v2.ColorJitter(brightness=0.1),
    v2.RandomPerspective(),
    v2.RandomAffine(degrees=20, translate=(0.1, 0.3), scale=(0.5, 0.75)),
    v2.RandomResizedCrop(args.input_size[1:], scale=(0.8, 1.0), ratio=(0.9, 1.1), antialias=True),
    v2.RandomInvert()
        ]
```

Figure 5: List of the augmentations applied to the images for computing the uncertainty via TTA.

## A.2 Time Complexity Analysis

Table 4 reports the average runtime per batch and the total runtime in seconds on CIFAR10. The increase in runtime from CBR (class-balanced, random) to BI (class-balanced, uncertainty-based) stems from the requirement to generate TTA images but remains fast in absolute terms. In fact, considering that in online-CL we are interested in updating the model every time a mini-batch of new data arrives, the average time for processing it remains reasonable with less than one second per batch. Since the batch size is the same for all datasets, this estimate reflects the average runtime per batch for all the considered datasets.

Table 4: Average runtime per batch and total runtime (in seconds) on CIFAR10.

|          | Runtime per batch | Total runtime |
|----------|-------------------|---------------|
| ER       | $\sim 0.22$s      | $\sim 42$s    |
| CBR      | $\sim 0.31$s      | $\sim 60$s    |
| BI (ours)| $\sim 0.69$s      | $\sim 132$s   |

## A.3 Memory Samples Analysis

In Figure 6, we analyse the composition and characteristics of the samples stored in the memory according to different strategies (random, top-k, and bottom-k). Figure 6a depicts the distribution of the confidence scores when samples are being stored in the memory. As one can see, although employing different strategies, the distributions look quite similar. Differently, in Figure 6b, the distributions of the BI-based uncertainty scores for the same samples differ significantly across the strategies. This shows the ability of BI to capture the epistemic uncertainty of the samples irrespective of the corresponding confidence score.

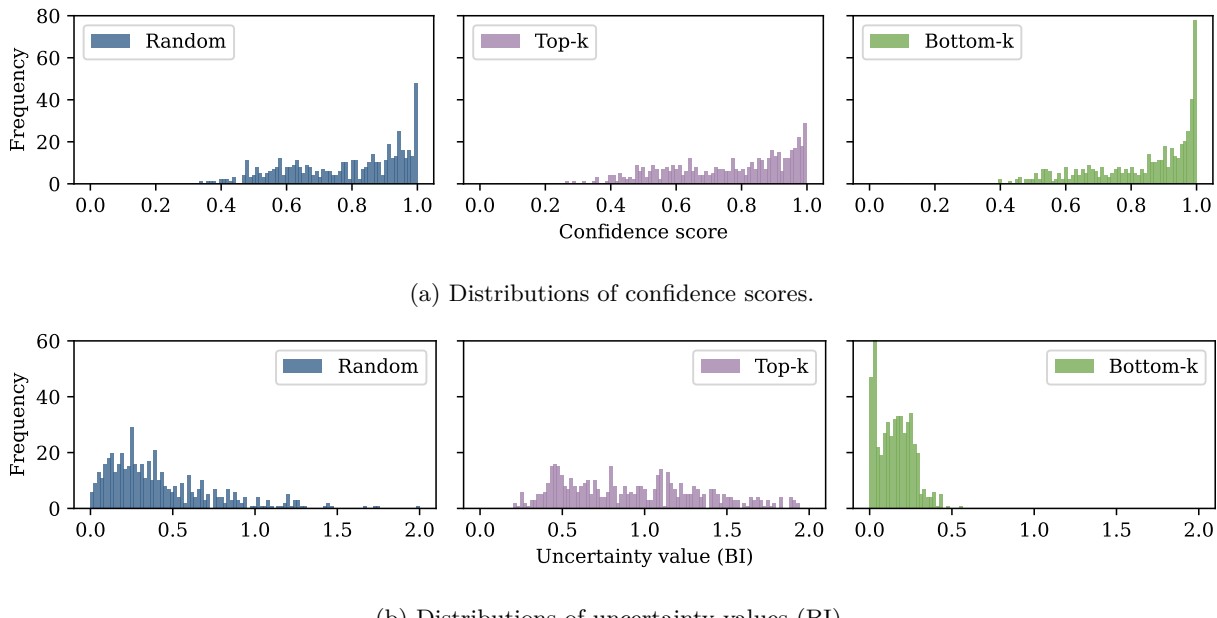

(a) Distributions of confidence scores.

(b) Distributions of uncertainty values (BI).

Figure 6: Distributions of confidence scores (a) and uncertainty values (b) for samples stored in the memory according to different strategies (random, top-k, and bottom-k).

## A.4 Relevance of Online-CL for AI-aided Medicine

In our experiments, we include BloodCell as an example of a realistic online-CL scenario that reflects a common challenge in AI-aided medicine: (i) classifying cells from blood smears is an important component in diagnostic hematology and (ii) its imbalanced nature reflects the common setting where different disease subtypes (that require differential treatment) occur with vastly different frequencies (class imbalance); with common diseases being common, hospitals tend to first see patients with the most common subtypes when collecting data for a ML-model and over time also patients with less common disease subtypes will present in the clinic. If the model is trained in the online-CL scenario it is not necessary to wait potentially a long time until sufficient patients of all subtypes have presented. Instead the model can be updated continually over time (online) whenever patients with a new subtype present. It is then crucial to mitigate CF e.g. via a memory based approach, to make sure all subtypes can be predicted well.

### A.5 Pseudocode of online-CL with BI

---

**Algorithm 1** Online-CL learning scheme

---

1: **Input** T: number of tasks, $bs$: batch size, $D_t$: task dataset.
2: **Notation** $\boldsymbol{\theta}$: model parameters, $\mathcal{M}$: memory buffer, $t$: current task, $b^t$ current batch of task $t$, $\boldsymbol{m}$: memory sample (size equal to $bs$).
3: **while** training not completed **do**
4:     $b^t \leftarrow \text{GetNextBatch}(D_t)$                                                             ▷ Get one batch from $t$
5:     **if** $t = 0$ **then**
6:         $\boldsymbol{\theta} \leftarrow \text{Train}(b^t)$
7:     **else**
8:         $\boldsymbol{m} \leftarrow \text{SampleFromMemory}(\mathcal{M})$                      ▷ Random sampling from memory
9:         $\boldsymbol{\theta} \leftarrow \text{Train}(b^t, \boldsymbol{m})$                                 ▷ Train with replay
10:     **end if**
11:     $\mathcal{M} \leftarrow \text{UpdateMemory}(\mathcal{M}, b^t)$
12: **end while**
13:
14: **function** $\text{UpdateMemory}(\mathcal{M}, b^t)$
15:     **for** $c \in b^t$ **do**                                              ▷ For each label $c$ in the current batch
16:         $\boldsymbol{m}_c \leftarrow \mathcal{M}[y = c]$                          ▷ Extract samples with label $c$ from memory
17:         $\boldsymbol{a}_c \leftarrow \boldsymbol{m}_c \cup b^t[y = c]$                         ▷ Get candidate samples for label $c$
18:         $\boldsymbol{u}_c \leftarrow u_{BI}(\text{TTA}(\boldsymbol{a}_c))$                      ▷ Compute uncertainty with Eq. (6)
19:         $\mathcal{M} \leftarrow \text{ReplaceSamples}(\boldsymbol{u}_c)$                      ▷ Update $\mathcal{M}$ based on $\boldsymbol{u}_c$
20:     **end for**
21:     **return** updated $\mathcal{M}$
22: **end function**

---

