# OpenReview forum: "How to Leverage Predictive Uncertainty Estimates for Reducing Catastrophic Forgetting in Online Continual Learning"
_TMLR — Accepted by TMLR_

### Review · Reviewer_3Hfb · 2024-12-09

**Summary Of Contributions:**

- This paper conducts an empirical study of different memory population strategies used in online continual learning. The empirical study considers popular image classification benchmark (CIFAR100) as well as a medical image classification task.
- These strategies revolve around different uncertainty metrics used for populating the replay memory. Bregman information is proposed as another uncertainty metric.
- Analysis is conducted on the qualities of the samples that should be included in the memory to alleviate catastrophic forgetting efficiently.

**Audience:**

Yes

**Broader Impact Concerns:**

While the paper includes some results on real-world inspired data, most of the experiments are conducted on small scale problems. Therefore, it does not seem like this work is necessarily going to have immediate impact on real world tasks.

**Claims And Evidence:**

Yes

**Requested Changes:**

- The introduction is hard to follow. Please update.
	- Motivate why the data can be processed only once.
	- Motivate why there is only limited memory available for replay.
	- Include a high-level definition of predictive uncertainty.
	- The first paragraph of the introduction is almost a full page long. Break it up for improved readability.
	- Fix grammar in the sentence "Furthermore, considering the quickly expanding landscape of recently rehearsal-based methods proposed, it is difficult to identify a clear strategy to exploit the memory at its best"
- Equation 1 comes before the "Preliminaries and Notation" section, yet it uses notation introduced in that section. Consider moving or removing the equation. If you keep it, please add the definitions of $c$, $\hat{c}$, and $\tilde{x}$ in the explanation for equation 1.
- Please include details for what model is used for computing the predicted score and logits in figure 2.
- There are many numbers in tables 1 and 2 but the discussion and analysis is lacking.
	- The error estimates in tables 1 and 2 are not explained. The bolding strategy is not explained. The step and bottom accuracies for BI seem to have error bars overlapping the mean. Some of the other methods, such as MS and LC also have error bars overlapping the mean of the bolded method.
	- What is the takeaway from the different ordering of the methods based on M?
- The BloodCell dataset it claimed to be realistic, yet very little details are provided. Please include a short description how the online continual learning setting is relevant to the realism.

**Strengths And Weaknesses:**

- The empirical study is systematic.
- The relevant related work is surveyed and discussed in detail.
- The presentation is somewhat hard to follow due to organizational issues and missing details.

---

> ### Author Response · Authors · 2024-12-20
> **Response to Reviewer 3Hfb**
>
> Thank you Reviewer 3Hfb for your thorough and helpful comments. Below are our responses to your requests.
>
> 1. **Introduction:** We restructured the introduction following your suggestions to make it clearer. As requested, we 1) added an explanation of the reasons behind the high-frequency updates and the limited memory available; 2) included a high-level definition of predictive uncertainty; 3) added line breaks to improve readability; and 4) fixed the grammar where needed.
>
> 2. **Equation 1 notation:** We added the definitions of the missing notations. Thanks for your suggestion.
>
> 3. **Table organisation and discussion:** Thanks for pointing this out. To facilitate easier comparison of different sampling strategies in Tables 1 and 2, we now highlight the columns for each strategy (top, step, and bottom) with different colors. Regarding your question about the different ordering of the methods based on $M$, we believe that, by increasing the memory size, the difference between the samples stored in the memory across methods is less pronounced. The difference in the selection method is more pronounced when the number of samples to store is smaller.
>
> 4. **BloodCell extended description:** Thank you for your suggestion. The revised version of the paper includes a more detailed description of the dataset and its practical relevance in an online CL setting. As now reported in the paper, this dataset represents a realistic online-CL scenario that reflects a common challenge in AI-aided medicine: (i) classifying cells from blood smears is an important component in diagnostic hematology and (ii) its imbalanced nature reflects the common setting where different disease subtypes (that require differential treatment) occur with vastly different frequencies (class imbalance); with common diseases being common, hospitals tend to first see patients with the most common subtypes when collecting data for a ML-model and over time also patients with less common disease subtypes will present in the clinic. If the model is trained in the online-CL scenario it is not necessary to wait potentially a long time until sufficient patients of all subtypes have presented. Instead, the model can be updated continually over time (online) whenever patients with a new subtype present. It is then crucial to mitigate CF e.g. via a memory-based approach, to make sure all subtypes can be predicted well.

---

### Review · Reviewer_pj6v · 2024-12-14

**Summary Of Contributions:**

The paper studies Online Continual Learning, with a focus on memory-based methods - approaches that maintain a fixed-size memory to enable replay such that the learned model is less likely to suffer from catastrophic forgetting. Such methods involve two key components: 1) memory population: decide which samples to include in the fixed-size memory 2) memory sampling: decide how to sample from the memory for the learning algorithm to replay.

The authors evaluate several existing memory population methods and propose a new method based on Bregman Information. The evaluation is conducted across several setups: 1) a more standard setup where CIFAR-10 and CIFAR-100 are divided into a sequence of tasks, with each task containing only a subset of classes, 2) an imbalanced setup using CIFAR-10 and CIFAR-100, but the number of samples for classes can be very different, and 3) a biomedical image dataset, BloodCell.

**Audience:**

Yes

**Claims And Evidence:**

No

**Requested Changes:**

Please refer to the section "Weakness" as well.

## More on Figure 1

Figure 1 is not properly explained. The labels are not defined until Section 4.2.2, and the meaning of “relative improvement” is not clarified. Additionally, the modification of a dataset to “LT” is introduced only in Section 5. Overall, this makes the placement of Figure 1 and the message it aims to convey very unclear.

## More on Figure 2

The term “predicted score” is not defined anywhere in the paper, and the label “Conf” is not explained. Moreover, the choice to represent the upper bar as “low” and the lower bar as “high” seems counterintuitive.

**Strengths And Weaknesses:**

# Strengths

The topic is highly relevant to the audience of TMLR. The paper evaluates several existing methods from the literature across diverse experimental setups and also introduces a new method. This work has the potential to provide valuable insights to the community.

# Weakness

One of the biggest weaknesses of this paper is its lack of clarity throughout the text. Specifically, 1) it excessively uses technical terms without clear explanations, 2) presents vague arguments and reasoning about the literature, and 3) fails to clearly differentiate between existing methods and the new method proposed in the paper. Another major issue is that the proposed method is weak in both its reasoning and the results.

While the clarity issues may seem minor individually, they accumulate and significantly impact the readability of the paper. I will highlight some of these issues in a general manner and provide a few examples, but I strongly urge the authors to carefully proofread and thoroughly revise the paper.

## 1. Terminology Issues

The paper contains a large number of terminologies, which can be acceptable if handled carefully. However, some terms feel unnecessary, and the essential ones are not introduced clearly, creating unnecessary obstacles for readers.

For instance, a quick search for words like “-based,” “approach,” “strategy,” and “technique” reveals a large number of technical terms. Some terms, like “retrieval strategy” and “update strategy,” are only used once, while others, such as “rehearsal” and “replay,” appear to be used interchangeably. While it’s acceptable to use synonyms, the lack of proper definitions for these terms can make the text overwhelming and confusing.

Another issue is the introduction of terms without sufficient explanation, as if assuming the readers are already familiar with them. For instance, “predictive uncertainty” is used extensively without clarification—predictive uncertainty of what? Uncertainty could refer to several things, such as the uncertainty of samples in the training set, the uncertainty of potentially unseen samples within the valid domain, etc. Additionally, is the uncertainty being discussed only irreducible data uncertainty (aleatoric uncertainty), or does it also include the reducible model uncertainty (epistemic uncertainty)? These terminology issues are significant because terms can have different meanings across domains. Additionally, explanations should avoid introducing further technical jargon, as this can create layers of nested terms without proper clarification.

## 2. Vague arguments and reasoning

I want to start with a general comment about the concept of “predictive uncertainty.” The paper does not provide a convincing explanation of why the effect of predictive uncertainty estimates on management is worth exploring, nor does it justify the approach of using Bregman Information (BI) being reasonable. To clarify, I am not opposed to exploring this direction; my concern is that the paper does not explain or motivate it well enough.

The paper also attempts to connect predictive uncertainty to several aspects, including 1) whether data points are “sitting close to the boundary” or “sitting at the center” 2) whether they are representative or not, and 3) whether they are easy-to-forget or easy-to-remember. However, these connections are presented in a disjointed and unclear manner, making the reasoning feel vague and unstructured.

## 3. Unclear differentiation between existing methods and the new method

The last paragraph of Section 2.2.1 does not successfully convey why using predictive uncertainty is an appealing approach. Additionally, the sentence “Different from their approaches, which …, our method exploits predictive uncertainty for the same purpose” is unclear in its implications. You dedicate several paragraphs to linking previous approaches to the idea of selecting samples based on uncertainty, but this sentence fails to clarify how your method differs from those approaches. Furthermore, your method, which is not described in detail at this point, does not appear to be presented as a more principled solution compared to the existing methods.

Similarly, in Section 4, you frequently claim that your approach differs from others but rarely describe how your method actually works. This lack of detail makes the distinctions between methods unclear. For example, you mention, “We introduce a class-balanced memory management … This is similar to Bang et al. (2021).” If they are similar, what exactly are the differences? Without clearer explanations, the novelty and uniqueness of your approach remain ambiguous.

## 4. The proposed method is weak in its idea and result

The authors state: “The goal of this work is not to propose a new method that outperforms the SOTA, but rather to present practical insights from an uncertainty-aware perspective on the desirable characteristics samples should have to alleviate CF in …” I want to emphasize that it is perfectly acceptable for the proposed method not to outperform state-of-the-art (SOTA) methods. However, a method is worth introducing only if the idea behind it follows strong reasoning—such that, before conducting experiments, we could reasonably expect it to achieve meaningful outcomes and potentially perform well.

In this paper, it remains unclear 1) what exactly is meant by “predictive uncertainty,” 2) why this concept is worth exploring, 3) why the interpretation of “uncertainty” is framed in the way described, and 4) why the use of Bregman Information (BI) was chosen as the foundation for the proposed method.

The proposed method, Bregman Information, is only mentioned sparsely throughout the paper, with the only explanation being a brief paragraph in Section 4.2.3. While some reasoning is provided earlier, it is not sufficiently clear. For example, the discussion of epistemic and aleatoric uncertainty includes the statement: "capture the aleatoric uncertainty which is inherent in the data, ..., the uncertainty about the data generating process (i.e., the epistemic uncertainty) is low." However, the "uncertainty about the data in the data-generating process" is precisely the aleatoric uncertainty inherent in the data. And hence, the reasoning that follows appears shaky.

## Questions and More Comments

### Section 1

In the statement, “Given the overlap between old and new information, the model tends to forget about the past knowledge to focus more on the newest tasks, …” I find the argument unclear.

Consider an extreme case where we are in a single-task scenario, and the samples are independently and identically distributed (i.i.d.) from a data-generating process, with the data arriving sequentially in batches. In this situation, the “overlap of the old and new information” would be expected to be substantial. However, the reasoning provided regarding forgetting past knowledge and focusing on new tasks does not seem to align with this scenario as described. The connection between overlap and forgetting requires further explanation and justification.

### Section 2

I find the argument about “forgetting” rather hand-wavy. The reasoning would be more convincing if it focused on whether the samples are close to the decision boundaries or not. Ultimately, what matters is the model’s ability to retain relevant information from the samples it has seen, but at the end of the process, all the samples are discarded.

In Section 2.2.2, you attempt to define the “uncertainty” of each sample by measuring how much it stays or shifts into another class after introducing noise. However, you do not specify how the labels for these perturbed samples are assigned. If you rely on classic data augmentation methods, the perturbed samples would inherit the same labels as the original ones, making the uncertainty of every sample effectively zero. If this is not the case, are the labels assigned by another fixed predictor?

You claim that the contribution of Eq. (1) on reducing CF is unclear, and you argue by saying how the “online” data was presented differently. However, Eq. (1) says nothing about how data comes. Therefore, I don’t understand your argument.

### Section 3

The notations are poorly defined and inconsistent. For example, n is used for the number of classes but later reused for a different purpose. Domains and codomains are not specified, b_t​ denotes a set while b_s​ represents a number, and terms like “class-balanced” and “high-frequency” are undefined. These issues make the paper hard to follow and need to be addressed.

### Section 4

Before Section 4, you repeatedly emphasize the importance of sample uncertainty in memory population, yet here you introduce a parallel discussion on maintaining class balance in the memory. If this aspect is central to your method, it should be discussed in the related works, or you should expand the related works section to include relevant literature.

For memory sampling, the descriptions provided feel arbitrary and lack focus. You do not specify any methods requiring sampling beyond uniform or i.i.d. sampling. As it stands, this discussion is more distracting than informative.

In Section 4.2, the description of logits is insufficient. Please explain more on what they represent and what you intend to do with them. Additionally, you do not specify how labels for perturbed samples are assigned. The variable “c” in your formulas is also undefined, making the meaning of these equations unclear. Please make sure the consistency of equations with Eq. (1), as it appears some formulas may be incorrect. It is also unclear whether you deviated Eq. (1) in an earlier section.

### Section 5

In Section 1, you mentioned there is a first part, where you will just use plain CIFAR-s and a second part, where you will use the -LT version of them. It’s better to make it clear also in the “Datasets” section.

In Section 5.1.2, introducing ER feels abrupt. Instead of “reservoir approach,” can you simply say data is randomly selected for memory and sampled randomly for replay? The mention of Dropout also lacks context and should be explained or removed.

The section needs thorough proofreading based on the earlier feedback.

---

> ### Author Response · Authors · 2024-12-20
> **Response to Reviewer pj6v (1/2)**
>
> Thank you Reviewer pj6v for your detailed and helpful feedback. Below are our responses to your concerns.
>
> 1,2. **Terminology issues, vague arguments and reasoning:** we modified the introduction to make sure each term is followed by its description. Furthermore, we restructured the introduction to connect in a clear way the research questions, the motivations of our investigation, the proposed method, and the hypothesis behind our proposal (see points below for further details).
>
> 3. **Unclear differentiation between existing methods and new method:** As we now explain in the revised version of the paper, the majority of the uncertainty estimates used in the online CL context mainly capture the aleatoric uncertainty (which is data-inherent and irreducible, i.e., cannot be minimized by gathering more data). In contrast, we hypothesize that focusing on epistemic uncertainty could give a benefit for reducing CF while maintaining similar predictive performance. This is because the epistemic uncertainty is model-centric and reducible and as such can for example be reduced by collecting more data in regions where the observed data is not representative of the true data generating distribution. We then build on the theoretical contribution from [1] who introduce a general bias-variance decomposition for strictly proper scores, and identify the Bregman Information as the variance term. We leverage this insight and, interpreting BI as a measure of epistemic uncertainty that is statistically well grounded in a bias-variance decomposition, are the first to suggest to use the Bregman Information for memory management. Although aleatoric and epistemic uncertainty measures appear similar in principle, their behaviour is different in practice (as also graphically explained in Figure 2 of the paper). Finally, to further improve the clarity of our proposed method, we reported in Algorithm 1 (Appendix A.5) the pseudocode of the online-CL learning scheme when using BI as an uncertainty estimate.
>
> 4. **Weak idea and results:** We have now reorganized the introduction to make sure to explain the reasoning and hypothesis behind our investigation and proposed methodology. Here, we report a summary of the reasoning steps behind our work. Our investigation starts from the following observation: there is conflicting evidence in the literature which type of data points (low uncertainty/most representative or high uncertainty/least representative) would reduce CF in a consistent manner in memory-based online-CL. Starting from this, we want to understand if the least or the most representative samples are more effective in combating CF and we analyze this problem from an uncertainty perspective. In particular, we focus on the uncertainty of the model in its prediction (also referred to as _predictive uncertainty_). However, predictive uncertainty can be seen as a composition of _aleatoric_ (data-inherent and irreducible) and _epistemic_ (model-centric and reducible by gathering more data) uncertainty, and different uncertainty scores focus more on one source or another. We observe that most of the common uncertainty scores employed in the literature (e.g., least confidence, entropy) mostly capture the irreducible _aleatoric_ uncertainty. In this paper, we hypothesize that for memory management, a better way to quantify if samples are representative or not is to focus on the reducible _epistemic_ uncertainty. For this reason, building on the Bregman decomposition from [1], we propose to use BI as an estimate of the _epistemic_ uncertainty. From the experimental results, we can draw the following conclusions: 1) the empirical results validate our hypothesis that using BI as measure of epistemic uncertainty delivers better results than using alternative uncertainty scores; 2) storing the most representative samples (i.e., the ones with the lowest uncertainty) and using them for replay helps the model in combating CF in a constistent manner. This result, summarised in Figure 1, answers the initial research question we wanted to investigate and can help practitioners understand the characteristics samples should have to alleviate CF in the online CL context. We made sure to rephrase any sentence containing the words 'data generating process' to avoid any misinterpretation.

---

> ### Author Response · Authors · 2024-12-20
> **Response to Reviewer pj6v (2/2)**
>
> 5. **Questions and more comments**
>
>    - Section 1: the statement that _"Given the overlap between old and new information, the model tends to forget about the past knowledge to focus more on the newest tasks"_ stems from the characteristics of deep neural networks when trained with non-stationary data streams (as in continual learning settings). In fact, incrementally updating the network with non-iid streams of data results in the so-called catastrophic forgetting which represents the inability of a network to perform well on previously seen data after updating with recent data [2].
>
>    - Section 2: as above, the argument about _forgetting_ is not our argument, but originates from the extensive literature in (online) continual learning and the characteristics of deep neural networks when trained with non-iid streams of data.
>
>    - Section 2.2.2: we follow the standard test-time-augmentation (TTA) algorithm where no label is required. Instead, we compare the similarity of predicted logit vectors under perturbations/augmentations and quantify how variable predictions around a certain point in the input space are. This variability quantifies the predictive uncertainty of the model under consideration and it is measured differently according to different uncertainty scores. In [1], the authors have demonstrated that a TTA-based estimator can reliably quantify BI.
>
>    - Section 3: thank you for pointing this out. The revised version of the paper includes an improved and more precise notation.
>
>    - Section 4: The class balance in the memory is important to consider also complex real-world use-cases where some classes may be underrepresented in the original dataset. However, this property of memory management can be decided in advance and does not represent a research direction by itself. Similarly, the strategy for sampling from the memory during replay can be chosen freely. Most of the related work employs random sampling since the memory is populated according to some specific strategy and random sampling is sufficient for the replay phase.
>
>    - Section 4.2: with logits, we refer to the unnormalized output of the last layer (classifier) of our learning model. These logits can be used to understand the uncertainty of our model in predictive terms. These numbers are used in different ways depending on the uncertainty score we employ. Thus, depending on the way the logits are used, different uncertainty scores deliver different results. We modified the notation of $c$ in the equations to $\hat{c}$ (which is now defined as the predicted class) to be consistent.
>
>    - Section 5: In the revised version of the paper, we highlighted in the 'Datasets' section that the LT datasets were used in the second part of our experiments.
>
> [1] Sebastian Gruber, Florian Buettner. "Uncertainty Estimates of Predictions via a General Bias-Variance Decomposition." Proceedings of The 26th International Conference on Artificial Intelligence and Statistics, PMLR 206:11331-11354, 2023.
>
> [2] Mai, Zheda, et al. "Online continual learning in image classification: An empirical survey." Neurocomputing 469 (2022): 28-51.

---

> > ### Comment · Reviewer_pj6v · 2025-01-09
> >
> > I appreciate the authors' efforts in revising the paper, and the introduction is indeed improved. However, several weaknesses persist.
> >
> > The argument for the proposed method remains unconvincing. For instance, on page 2, you question how “representative samples” relate to combating CF and hypothesize using epistemic uncertainty. However, representativeness is inherently a property of the data and the task, not the model. While predictive uncertainty involves the model, it introduces an additional layer of uncertainty based on the chosen measure, rather than solely reflecting a property of the data. Moreover, from a statistical learning perspective, epistemic uncertainty typically decreases with more data in a task. In the continual learning setting, where tasks are distinct and we do not revisit earlier tasks to add data, there is limited theoretical basis to predict how epistemic uncertainty will behave. Additionally, the fact that prior methods focused on aleatoric uncertainty does not directly justify focusing on epistemic uncertainty, as novelty alone is insufficient. Furthermore, the decision to focus exclusively on epistemic uncertainty rather than considering both epistemic and aleatoric uncertainties lacks sufficient justification.
> >
> > The results presented in the tables also show limited improvement over existing methods.
> >
> > Lastly, notations remain unclear. For example, in Section 4.2.2, \hat{c} appears without explanation. While \hat{c} in Eq. (1) is introduced as a factor being optimized, its role in Section 4.2.2 is undefined and creates confusion.

---

> > > ### Author Response · Authors · 2025-01-10
> > > **Response to official comment by Reviewer pj6v**
> > >
> > > Thank you for the feedback. Here is the response to the remaining concerns:
> > >
> > > We would first like to address a potential misunderstanding: we do not "focus exclusively on epistemic uncertainty"—we comprehensively assess a wide range of uncertainty measures, quantifying both epistemic and aleatoric uncertainty. A major contribution of our work is that we demonstrate that epistemic uncertainty is most beneficial for combating catastrophic forgetting (see, for example, Fig. 1).
> > >
> > > The use of epistemic uncertainty to evaluate data representativeness is motivated by the behaviour of epistemic uncertainty itself. Measures of aleatoric uncertainty have high confidence in areas that are distant from the decision boundary. As such, samples with high confidence from an aleatoric perspective reflect data points that are distant from the decision boundary irrespective of the fact that they may be outliers or not. In the epistemic case, instead, samples with high confidence (low epistemic uncertainty) indicate that the observed data sufficiently support the inferred patterns - this can be interpreted as samples being representative of the corresponding data distribution; it is in contrast to areas with high epistemic uncertainty, where adding more data points would decrease uncertainty as the observed data in those areas is not representative (i.e. does not sufficiently support inferred patterns). This is shown in Figure 2 where the confidence of the Bregman Information is low in areas where the data evidence is not sufficient. This key insight motivates us to suggest using measures of epistemic uncertainty like BI. \
> > > We have now clarified in the manuscript what we mean by the term "representative samples". We clarify that while representativeness is indeed a property of the data, we refer to it as a property that can be inferred from a model via (low) epistemic uncertainty. We now explicitly state that this is an interpretation of epistemic uncertainty. We do not wish to imply that representativeness is the property of a model, but rather indicates whether the observed data sufficiently support the patterns inferred by the model.
> > >
> > > Our goal is to create a representative (in the above interpretation) and informative memory of the current task _t_ for the future. When training on a certain task _t_, we are indeed adding new evidence to task _t_ every time we receive a new batch of data of it. We want to use this information to populate the memory with the most representative samples of the current task. However, as explained above, the memory will be filled with different samples (with different characteristics) when using measures of aleatoric or epistemic uncertainty. We believe that the difference in the memory composition when using BI compared to other strategies affects the capacity of the learning model to remember the past information (leading to a higher or lower forgetting).
> > >
> > > The results corroborate our hypothesis. Considering the results reported in the 'top' columns (in blue) and the 'bottom' columns (in green) it is possible to observe a clear pattern: the bottom strategy reduces consistently CF compared to the top strategy. Furthermore, if we inspect the 'bottom' columns only, we can observe that our approach consistently outperforms the other strategies in terms of CF. This evidence is also summarised graphically in Figure 1 where 1) dark bars (least uncertain) are consistently higher than light bars (most uncertain) and also consistently above ER (better than ER in terms of CF), and 2) the dark green bars (BI) are higher than the dark blue bars (other uncertainty scores).
> > >
> > > Thus, the results are aligned with the scope of the paper and:
> > >
> > > 1. clarify that the most representative samples (i.e. those with low epistemic uncertainty) are the most effective at mitigating CF;
> > > 2. support our hypothesis that BI, given its emphasis on epistemic uncertainty, is better suited for reducing CF in comparison with well-established uncertainty scores (which mostly capture the irreducible aleatoric uncertainty).
> > >
> > > Finally, we would like to kindly emphasize that $\hat{c}$ has the same meaning in all the equations. As a matter of fact, in Eq. 1, it is not a factor being optimised.

---

### Review · Reviewer_PuAH · 2024-12-16

**Summary Of Contributions:**

This paper investigates the problem of catastrophic forgetting in online continual learning (CL) using uncertainty estimation. It evaluates several predictive uncertainty metrics for classification loss functions, with a focus on Bregman Information (BI). The authors propose that selecting the most representative samples—characterized by low uncertainty according to BI—is an effective strategy to reduce catastrophic forgetting. The findings are tested on the experiments of class-balanced and imbalanced datasets.

**Audience:**

Yes

**Broader Impact Concerns:**

No broader impact concerns.

**Claims And Evidence:**

Yes

**Requested Changes:**

1. Expand the discussion to include [1] (A-GEM) and [2] (Bioslam) and conduct experiments comparing the proposed method with these baselines.
2. Reassess the claim about the effectiveness of least uncertain samples in reducing CF and make it more precise.
3. Include an analysis of the uncertainty spectrum of samples stored in the memory buffer and compare it to random sampling (ER) to provide deeper insights into the proposed approach.
4. Provide time and memory complexity analysis for the proposed method and compare it with ER.

**Strengths And Weaknesses:**

[Strengths]
1. The paper studies a critical challenge in online continual learning—catastrophic forgetting—by leveraging uncertainty estimation, which is both novel and applicable to dynamic real-world environments.

2.  Applying BI-based uncertainty as a method for selecting representative samples is a theoretically grounded and practical approach.

3. Experiments on imbalanced datasets replicate real-world scenarios, showcasing the practical utility of the proposed approach.

[Weaknesses]
 1. This paper does not thoroughly address related works. For instance, it is crucial to evaluate the proposed method against Average Gradient Episodic Memory (A-GEM) [1]. Additionally, the study overlooks relevant work such as Bioslam [2], which also explores memory-replay-based continual learning. Bioslam incorporates an internal reward mechanism (Equation 12) to guide sample selection, which aligns with the concept of uncertainty estimation.

2. The claim that "least uncertain samples are the most effective at mitigating CF" is not rigorously supported. For example, random sampling (ER) achieves competitive or better results in some cases (e.g., Table 2). This claim should be rephrased with appropriate conditions or clarified to emphasize relative effectiveness of least uncertain samples over most uncertain samples.

3. The paper would benefit from analyzing the uncertainty spectrum of samples stored in the buffer. For instance, it should investigate whether most samples belong to low-uncertainty regions and compare the uncertainty distributions of ER (random sampling)  and the proposed method.

4. While the proposed method involves additional computation for uncertainty estimation, the paper does not provide a comparison of time or memory complexity with simpler methods like ER.

[1]  A. Chaudhry, et al. “Efficient lifelong learning with a-gem. In International Conference on Learning Representations,” 2018.
[2] Yin, Peng, et al. "Bioslam: A bioinspired lifelong memory system for general place recognition." IEEE Transactions on Robotics (2023).

---

> ### Author Response · Authors · 2024-12-20
> **Response to Reviewer PuAH**
>
> We thank Reviewer PuAH for the constructive and valuable comments. Below are our responses to your concerns and questions.
>
> 1. **Additional related works:** Thank you for pointing out A-GEM [1] and BioSLAM [2]. A-GEM presents a gradient-based approach for online-CL that computes the gradient of the current mini-batch and compares it with the gradient of a randomly sampled set of the same size from the memory (which is updated randomly). If the dot product between the current gradient and the memory gradient is negative, the current gradient is projected. Otherwise, the gradient is used normally. We now include an additional comparison with A-GEM in Tables 1, 2, and 3. From the results, this gradient-based approach is not able to provide competitive results compared with uncertainty-based approaches or ER. \
> BioSLAM, instead, presents a lifelong approach for Simultaneous Localization and Mapping (SLAM) for continuous localization and incremental place recognition in robotics. Although the approach presents some similarities in the nature of the setting (lifelong), the approach is tailored for SLAM. In particular, it employs a dual-memory selection mechanism with static and dynamic memory zones and a sleeping cycle for memory consolidation that would require substantial adjustments to work in the general online CL setting. Furthermore, the memory system also includes a place feature encoding procedure for new observations and it is not clear how to generalise that for non-SLAM use cases. For this reason, we argue that a comparison to BioSLAM is out of the scope of this work and instead focussed on including A-GEM as an additional baseline.
>
> 3. **Claim reformulation:** It is true that, in some cases, ER achieves competitive or better results than our 'bottom-k' strategy in terms of accuracy. However, we do believe that our statement that "least uncertain samples are the most effective at mitigating CF" holds true. To facilitate a more straightforward comparison of strategies (least uncertain/most uncertain), we now highlight the columns corresponding to each strategy (top, step, and bottom) with different colors. If we consider the results reported in the 'top' columns (in blue) and the 'bottom' columns (in green) is it possible to observe a clear pattern: the bottom strategy reduces consistently CF compared to the top strategy. This is also summarised graphically in Figure 1 where dark bars (least uncertain) are consistently higher than light bars (most uncertain) and also consistently above ER (better than ER in terms of CF). Furthermore, even in comparison with ER, we can notice that, in the few cases where the accuracy of ER is relatively higher, the forgetting is also significantly higher than the 'bottom' strategy. This further corroborates our claim that the least uncertain samples are the most effective in reducing catastrophic forgetting.
>
> 4. **Uncertainty spectrum analysis:** Thank you for suggesting this additional analysis. We now include two additional figures to depict the distributions of the uncertainty values and of the confidence scores (Figure 6, Appendix A.3) for ER, bottom, and top strategy. The plots nicely represent the characteristics of the samples stored in the memory and the difference between the confidence score and the uncertainty score distributions. This shows the ability of BI to capture the epistemic uncertainty of the samples irrespective of the corresponding confidence scores.
>
> 5. **Time complexity analysis:** We now provide a time complexity analysis of the proposed uncertainty-based approach with ER and CBR in Table 4. The table reports the average runtime per batch and the total runtime in seconds on CIFAR10. The increase in runtime from CBR (class-balanced, random) to BI (class-balanced, uncertainty-based) stems from the requirement to generate TTA images but remains fast in absolute terms. In fact, considering that in online-CL we are interested in updating the model every time a mini-batch of new data arrives, the time for processing it remains reasonable with less than one second per batch (0.22s for ER, 0.69s for BI). Since the batch size is the same for all datasets (and they all have similar image sizes), this estimate reflects the average runtime per batch for all the considered datasets.
>
> [1] A. Chaudhry, et al. “Efficient lifelong learning with a-gem. In International Conference on Learning Representations,” 2018.
>
> [2] Yin, Peng, et al. "Bioslam: A bioinspired lifelong memory system for general place recognition." IEEE Transactions on Robotics (2023).

---

### Author Response · Authors · 2024-12-20
**General response to reviewers**

We thank the reviewers for their detailed and thoughtful feedback. We are pleased that the reviewers appreciate our systematic empirical study (reviewer 3Hfb) that compares our proposed BI-based memory population strategy (which is theoretically grounded and practical – reviewer PuAH) with other uncertainty scores and showcases the practical utility of the proposed approach on imbalanced datasets (reviewer PuAH). The topic is deemed as ""highly relevant" and "has the potential to provide valuable insights to the community" (reviewer pj6v).

We are thankful for the encouraging and constructive comments aimed at improving the quality and soundness of our manuscript. Below we summarize the key points addressed during the rebuttal which are also reflected in the updated version of the paper (changes from the original submission are highlighted in blue):

- **Introduction**: in response to reviewers pj6v and 3Hfb, we reorganized the introduction to make sure to explain the reasoning and hypothesis behind our investigation and proposed methodology, and to make sure each term is followed by its description. Furthermore, we improved the overall readability of the section.
- **Additional experiments and analysis**: as requested by reviewer PuAH, we included 1) additional experiments with A-GEM, 2) an analysis of the uncertainty spectrum of the samples stored in the memory buffer, and 3) a time complexity analysis between ER and the proposed method.
- **Improved notation and descriptions**: as pointed out by reviewers 3Hfb and pj6v, we improved the notation throughout the whole manuscript and the description of terms and datasets.
- **Improved overall presentation**: we rephrased and restructured different sections of the manuscript to improve the presentation. To facilitate a more straightforward comparison of strategies (least uncertain/most uncertain) in Tables 1 and 2, we now highlight the columns corresponding to each strategy (top, step, and bottom) with different colors.

---

### Decision · Action_Editor_gAYR · 2025-02-06

**Recommendation:** Accept as is

**Comment:**

While the initial submission had some important flaws, especially around some technical language, it seems that those have mostly been addressed. Overall the work is an interesting scientific contribution. Even though the novelty and immediate impact of this work are not particularly high, the work is correct and belongs in TMLR.

**Audience:**

Yes, this work is relevant.

**Claims And Evidence:**

To me this paper passes the TMLR bar: it addresses a relevant topic, it has a clearly stated hypothesis which is supported by experiments. Although the writing of the paper was not entirely well received, the authors have worked to improve it and create an accessible contribution.